# Regional Differences in the Prevalence of Anaemia and Associated Risk Factors among Infants Aged 0–23 Months in China: China Nutrition and Health Surveillance

**DOI:** 10.3390/nu13041293

**Published:** 2021-04-14

**Authors:** Shujuan Li, Yacong Bo, Hongyan Ren, Chen Zhou, Xiangqian Lao, Liyun Zhao, Dongmei Yu

**Affiliations:** 1National Institute for Nutrition and Health, Chinese Center for Disease Control and Prevention, Beijing 100050, China; lisj@ninh.chinacdc.cn (S.L.); zc983702782@163.com (C.Z.); zhaoly@ninh.chinacdc.cn (L.Z.); 2State Key Laboratory of Resources and Environmental Information System, Institute of Geographic Sciences and Natural Resources Research, Chinese Academy of Sciences, Beijing 100101, China; renhy@lreis.ac.cn; 3Jockey Club School of Public Health and Primary Care, the Chinese University of Hong Kong, Hong Kong SAR 999077, China; boyacong777@gmail.com (Y.B.); xqlao@cuhk.edu.hk (X.L.)

**Keywords:** anaemia, 0–23-month-old infants, regional differences, risk factors, China

## Abstract

Infantile anaemia has been a severe public health problem in China for decades. However, it is unclear whether there are regional differences in the prevalence of anaemia. In this study, we used data from the China Nutrition and Health Surveillance (CNHS) to assess the prevalence of anaemia and the risk factors associated with its prevalence in different regions. We included 9596 infants aged 0–23 months from the CNHS 2013 database. An infant was diagnosed with anaemia if he/she had a haemoglobin concentration of <110 g/L. We used multivariate logistic regression to investigate the potential risk factors associated with the development of anaemia. We found that anaemia was present in 2126 (22.15%) of the infants assessed. Approximately 95% of these cases were classified as mild anaemia. Based on the guidelines laid out by the World Health Organization, 5.5% and 43.6% of the surveillance sites were categorized as having severe and moderate epidemic levels of anaemia, respectively. The prevalence of infantile anaemia in Eastern, Central and Western China was 16.67%, 22.25% and 27.44%, respectively. Premature birth, low birth weight, breastfeeding and residence in Western China were significantly associated with higher odds of developing anaemia. Female sex and having mothers with high levels of education and maternal birth age >25 years were associated with lower odds of developing anaemia. In conclusion, we observed significant regional disparities in the prevalence of infantile anaemia in China. Western China had the highest prevalence of infantile anaemia, and rural regions showed a higher prevalence of anaemia than urban regions.

## 1. Introduction

Anaemia, which affects approximately 1.62 billion (24.8%) people worldwide, is a global public health problem. It can occur at any stage of life but is more prevalent in children aged < 5 years and especially in children aged < 2 years [1]. Notably, during the first 1000 days of life, the brain is particularly vulnerable to psychosocial, environmental and biological factors. This phase is critical to the healthy development of children [2]. As such, the adverse effects of anaemia on brain development during this period are irreversible, even if anaemia is corrected during later stages of childhood [3]. 

The prevalence of infantile anaemia has been examined in some regions of China [4,5,6], but at relatively small scales, and there are few data that are representative of the entire country. In addition, the prevalence of childhood anaemia may differ with region, maternal anaemia status and socioeconomic status [7]. Few studies have evaluated regional disparities in infantile anaemia prevalence, especially between urban and rural areas. As infants under two years of age grow and develop rapidly, an age difference of mere months may affect their degree of anaemia and other associated factors. These associations need to be investigated in further detail.

The National Nutrition Plan (2017–2030) of China mandates that the anaemia rate in children under five years be reduced to <10% by 2030. To achieve this goal, it is necessary to map the prevalence of infantile anaemia across the country and identify areas of high anaemia prevalence. In this study, we analysed infant data from the China Nutrition and Health Surveillance (CNHS) database to determine the national and regional prevalence of infantile anaemia, based on the measurement of infants’ haemoglobin (Hb) concentrations. We also explored the potential factors associated with a high prevalence of infantile anaemia. 

## 2. Materials and Methods 

### 2.1. Study Design and Participants

The CNHS (2010–2013) was a national cross-sectional study with representative participants covering 31 provinces, municipalities as well as autonomous regions of China. It was conducted by the National Institute for Nutrition and Health, the Chinese Centre for Disease Control and Prevention. In 2013, a multistage stratified cluster sampling method was used to investigate the nutritional status of children aged 0–5 years and that of lactating mothers across the entire country. Fifty-five surveillance sites (cities/districts/counties) were selected, based on the principle of representation [8]; that is, three towns and three village-neighbourhood committees were selected randomly from each surveillance site. If the population size of the selected neighbourhood committee was less than the required sample size, it was increased as required. All of the children’s parents (or other caregivers) were asked to sign an informed consent form prior to joining the study. The protocol for this study was approved by the Ethics Committee of the National Institute for Nutrition and Food Safety of the Chinese Centre for Disease Control and Prevention (No. 2013-018).

As shown in Figure 1, 11,779 infants aged 0–23 months for whom Hb concentration measurements were available were initially included. We excluded 2,183 infants, as they were missing information on maternal characteristics (*n* = 497) or feeding status (*n* = 1686). Finally, 9596 infants were included in our analysis. Of these, 2959 infants were aged 0–5 months, 3185 were aged 6–11 months and 3452 were aged 12–23 months.

### 2.2. Data Collection

The CNHS adopted a structured questionnaire to collect detailed information on a broad range of factors. The following information was used in the present study: demographic factors (infant’s age, sex, ethnicity, mother’s age), socioeconomic status (mother’s level of education, family income and type of toilet used in the household), status of premature birth, infant’s birth weight and infant’s feeding status (whether breastfeeding and if so, the duration of breastfeeding).

A diet survey was conducted to collect information on the infants’ minimum dietary diversity (MDD). Based on the guidelines laid out by the World Health Organization (WHO), an infant was defined as meeting the MDD if he/she received four or more groups from the following seven groups of food: (1) cereals and rhizomes, (2) beans, (3) milk and dairy products, (4) meat (animal liver and meat), (5) eggs, (6) fruits and vegetables rich in vitamin A (food that is yellow on the inside, and/or any dark green leafy vegetables) and (7) any other vegetables and fruits. We also collected information on whether the infant received any iron-rich foods or iron supplements in the 24 h preceding the survey.

We collected blood samples from the fingers of the infants and evaluated the Hb concentration using the cyanide high-iron method [9]. Infants living at altitudes < 1000 m above sea level were diagnosed with anaemia when their Hb concentration was <110 g/L. Infants with a Hb concentration of 90–110 g/L were diagnosed as having mild anaemia and those with a Hb concentration of <90 g/L were diagnosed as having moderate or severe anaemia. For those living at altitudes > 1000 m above sea level, the thresholds were adjusted based on the following formula:

Hb adjustment = −0.032 × (altitude × 0.0032808) + 0.022 × (altitude × 0.0032808) [10].

### 2.3. Statistical Analysis

SAS v9.4 software was used for statistical analyses. Hb level was calculated with the “Surveymeans” procedure and tested with the Wilcoxon test, and anaemia prevalence was calculated with the “Surveyfreq” procedure. We first calculated the overall and sex-specific concentrations of Hb and the prevalence of anaemia. We then classified the participants as rural or urban residents, based on the code for statistical division and code for urban rural division in 2012 from the National Bureau of Statistics of China [11], and calculated the rural and urban Hb concentrations and anaemia prevalence. Finally, we grouped the participants into three groups according to geographical region, namely Eastern, Central and Western regions (Figure 2A), in accordance with the China Health Statistical Yearbook [12]. The corresponding concentrations of Hb and prevalence of anaemia were then calculated for each region.

We used a multivariate logistic regression model to explore the potential risk factors associated with the prevalence of anaemia. A stepwise method of variables entry was used. We conducted a literature review of the published literature pertaining to potential predictors/risk factors associated with the development of anaemia and considered the following factors: age (months), sex (male vs. female); ethnicity (Han Chinese vs. non-Han Chinese); premature birth (yes vs. no); low birth weight (LBW; yes vs. no); maternal age (≤25 years vs. >25 years); MDD (yes vs. no); intake of iron-rich food or iron supplements (yes vs. no); level of mother’s education (>secondary school vs. <secondary school); family income (<RMB35,000 vs. ≥RMB35,000); toilet type (sanitary vs. unsanitary; sanitary toilet refers to one with water-flushing facilities and/or a dry toilet), regional socioeconomic status, as defined by the central government big city (BC) referred to the central urban areas with a population of more than 1 million municipalities directly under the central government, cities under separate planning and provincial capital cities; small or medium-sized city (SMC) referred to all the districts and county-level cities outside the central urban areas of the above-mentioned big cities and the county-level cities or districts among the 592 poverty-stricken counties; poverty-stricken county (PC), county-level cities or districts are removed from the 592 counties identified in the “2001–2010 national program for poverty alleviation and development in rural areas”; or ordinary county (OC), counties except PC); residential area (urban vs. rural); and geographical region (Eastern, Central and Western). A two-tailed *p*-value of *p* < 0.05 was considered to indicate statistical significance.

## 3. Results

### 3.1. General Characteristics of the Participants

Table 1 summarises the characteristics of the included infants and their mothers. Of the 9596 infants, slightly more than half (51.28%) were male. Most of the infants belonged to the Han ethnic group (86.08%) and were term births (89.08%). Most of the mothers had an educational level of high school or lower and were aged 25 years or older at the birth of their infants.

### 3.2. Hb and Anaemia Status

#### 3.2.1. Hb Status

As shown in Table 2, the mean Hb concentrations were 120.44 ± 0.77 g/L, 120.96 ± 0.84 g/L, 120.23 ± 0.73 g/L and 120.97 ± 1.22 g/L for male infants, female infants, infants living in urban areas and those living in rural areas, respectively. In addition, the Hb concentrations were slightly higher in infants aged 12–23 months across all subgroups.

The spatial distribution of the 55 selected surveillance sites is depicted in Figure 2A. Due to the relatively small population in the Western region, we monitored fewer surveillance sites in this region than in the Eastern region. The Hb concentrations of infants in the Central region were slightly higher than those in the Western and Eastern regions (Figure 2B).

#### 3.2.2. Prevalence of Anaemia in Infants

In 2013, the overall anaemia prevalence in Chinese infants aged 0–23 months was 22.15%, with 95.3% of the affected infants having mild anaemia and 4.7% having moderate anaemia. There were no recorded cases of severe anaemia. Anaemia was more prevalent in male infants (23.14%) than in female infants (20.96%). The prevalence of anaemia among infants living in rural areas (23.57%) was higher than that in infants living in urban areas (20.02%). Similarly, infants aged 6–11 months (30.64%) were more likely to have anaemia than those aged 0–5 months (25.30%) or 12–23 months (16.97%). Moreover, the prevalence of anaemia in the Eastern region (16.67%; 95% confidence interval (CI): 15.50–17.84) was significantly lower than that in the Central (22.25%; 95% CI: 20.88–23.63) and Western regions (27.44%; 95% CI: 25.93–28.95). Further details are given in Table 3.

The spatial distribution of the prevalence of anaemia is presented in Figure 2C–F. The WHO classifies a site with an anaemia detection rate of ≥40% as having a severe epidemic, a site with a rate of 20.0–39.9% as having a moderate epidemic, a site with a rate of 5.0–19.9% as having a mild epidemic and a site with a rate of ≤4.9% as normal [13]. Thus, of the 55 surveillance sites selected for this study, 3 had a severe epidemic, 24 had a moderate epidemic, 27 had a mild epidemic and 1 was under a normal epidemic level. The prevalence of anaemia across all 55 sites ranged from 4.58 to 56.70%, with the highest prevalence in the Western region, followed by those in the Central and Eastern regions (Figure 2C). Figure 2D–F shows the prevalence of anaemia stratified by the age groups.

### 3.3. Factors Associated with the Prevalence of Anaemia in Infants Aged 0–23 Months

Table 4 shows the factors associated with the prevalence of anaemia in infants aged 0–23 months, derived from multivariate logistic regression analyses. For infants aged <6 months, female sex (odds ratio (OR) = 0.80; 95% CI: 0.68–0.95; *p* = 0.01) and having a mother with a higher level of education (OR = 0.77; 95% CI: 0.63–0.95; *p* = 0.01) were associated with lower odds of developing anaemia. In contrast, premature birth (OR = 1.38; 95% CI: 1.06–1.81; *p* = 0.02), LBW (OR = 1.99; 95% CI: 1.28–3.09; *p* = 0.00), breastfeeding (OR = 1.34; 95% CI: 1.04–1.72; *p* = 0.02) and residing in the Western region of China (OR = 1.66; 95% CI: 1.35–2.04; *p* < 0.01) were associated with higher odds of developing anaemia.

For infants aged 6–11 months, female sex (OR = 0.77; 95% CI: 0.66–0.90; *p* = 0.00) and having a mother with a higher level of education (OR = 0.58; 95% CI: 0.47–0.72; *p* < 0.01) were associated with lower odds of developing anaemia. As in the previous age group, premature birth (OR = 1.30; 95% CI: 1.01–1.67; *p* = 0.04), breastfeeding (OR = 2.18; 95% CI: 1.84–2.60; *p* < 0.01) and residing in the Western region of China (OR = 1.95; 95% CI: 1.60–2.37; *p* < 0.01) were associated with higher odds of developing anaemia.

For infants aged 12–23 months, individuals who met the MDD (OR = 0.60; 95% CI: 0.49–0.73; *p* < 0.01), those whose mother had a higher level of education (OR = 0.70; 95% CI: 0.51–0.95; *p* = 0.02) and those whose mother had a maternal birth age of >25 years (OR = 0.70; 95% CI: 0.57–0.85; *p* < 0.01) had lower odds of developing anaemia. In contrast, breastfeeding (OR = 2.17; 95% CI: 1.75–2.69; *p* < 0.01), residing in small or middle-sized cities (OR = 1.80; 95% CI: 1.30–2.50; *p* < 0.01) and living in the Central region of China (OR = 1.63; 95% CI: 1.25–2.13; *p* < 0.01), living in the West region of China (OR = 2.11; 95% CI: 1.62–2.76; *p* < 0.01)were associated with higher odds of developing anaemia.

## 4. Discussion

In this study, we determined the prevalence and distribution of anaemia in a nationally representative sample of Chinese infants aged 0–23 months. We found a substantial prevalence of anaemia among all infants aged 0–23 months: 22.15% were diagnosed with anaemia, of whom approximately 95.3% had mild anaemia, and 4.7% had moderate anaemia. The prevalence of anaemia among infants aged 0–23 months varied widely across provinces. Anaemia was more commonly seen in infants aged 6–11 months and in those living in the Western region. We also found that the risk factors associated with the prevalence of infantile anaemia were sex, maternal age, maternal educational level, premature birth, birth weight, breastfeeding and meeting MDD.

### 4.1. Burden of Infantile Anaemia in China in 2013

The prevalence of infantile anaemia observed in our study was lower than that among preschool children worldwide (47.4%) [14], in Latin America and the Caribbean (32.9%) [15] and in Africa (50.4–70.9%) [16,17] but higher than that among children aged 6–59 months in Canada (9.4%) and in the USA in 2013 (6.8%) [18]. The WHO estimates that approximately 1.62 billion people worldwide suffer from anaemia, of whom 293 million are children of preschool age [14]. A previous study in the Shanxi Province of China reported that approximately 54.3% of infants aged 6–11 months had anaemia and that 24.3% of infants in rural China had moderate or severe anaemia [19]. In addition, it revealed that the prevalence of anaemia was highest among infants aged 6–11 months, an age at which crucial developments in psychomotor skills occur.

### 4.2. Distribution Map of Infantile Anaemia in China in 2013

Based on the WHO classification system for anaemia prevalence [20], it is apparent that anaemia remains a severe public health problem in China. Approximately half of the study sites (27 out of 55 surveillance sites) had cases of moderate to severe anaemia among infants aged 0–23 months. The map of anaemia prevalence suggests that the prevalence of infantile anaemia is highest in the Western region, followed by the Central and Eastern regions. Our estimation of infantile anaemia prevalence at the regional and country level produced an important cartographic resource and provides important new evidence regarding sub-country-level priority areas for anaemia control.

### 4.3. Factors Associated with the Prevalence of Anaemia in Chinese Infants Aged 0–23 Months

Previous studies have demonstrated that socioeconomic status is a risk factor associated with the development of anaemia. Our study found that a higher level of maternal education and residence in economically developed areas were associated with lower odds of developing anaemia. This observation is in concordance with findings from Ghana [21], Guinea [22], India [23,24] and other studies in China [25]. Although the socioeconomic indicators for China (e.g., food expenditure, average disposable income and gross domestic product per capita) have increased in both urban and rural areas over the past decades [26,27], holistic socioeconomic conditions remain better in urban and Eastern/Central areas than in rural and Western areas [28]. The prevalence of infantile anaemia therefore remains higher in rural areas than in urban areas, although this difference is decreasing. Improving socioeconomic status may thus play an important role in reducing the prevalence of anaemia.

Many studies have demonstrated that maternal and reproductive health (e.g., premature birth, LBW, low maternal birth age) are associated with a higher risk of developing childhood anaemia [23,29,30]. Consistent with these results, our study showed that these risk factors were strongly associated with higher odds of developing anaemia among Chinese infants aged 0–23 months. This might be because infants with LBW are born with reduced iron stores [7], which play a pivotal role in the incidence of anaemia. In addition, younger mothers have less experience in caring for and rearing children than older mothers.

Our study also revealed that infant feeding is another important factor that is associated with the burden of infantile anaemia in China. We found that breast-fed infants were more likely to suffer from anaemia (OR: 1.34–2.18, *p* < 0.02), which is consistent with a previous study of Chinese infants [25]. This can be ascribed to the specific characteristics of infants aged 6–11 months. After six months of age, the iron provided by breast milk is no longer sufficient to support the needs of infants. At this stage, supplementary food not given in a timely or reasonable manner may lead to iron deficiency and anaemia. Food supplementation is an important factor associated with the development of infantile anaemia, as reflected by the MDD. We found that meeting the MDD guidelines was associated with low odds of developing anaemia among infants aged 12–23 months, which is consistent with a previous study conducted in India [31].

According to a study, >55% of children in East Asia have iron-deficiency anaemia (IDA) [32], and the proportion in China can be as high as >80% [4]. Only if the anaemia was IDA were iron interventions effective in preventing anaemia [33]. Efforts should be made to reduce the prevalence of infant anaemia in China. Infant feeding is a key point to solving iron-deficiency anaemia personally. Scientific and reasonable complementary food addition include in time food addition and guarantee a variety of food. To the eliminate gap in different areas, complementary food fortification such as Ying Yang Bao (YYB) should implement in more rural areas and cover more children under 23 months because the YYB project greatly reduced the anaemia of children in poverty-stricken rural countries [34].

### 4.4. Strengths and Limitations

The strengths of this study lie in its large representative sample of Chinese infants. The CNHS (2010–2013) was a national surveillance and was implemented by National Institute for Nutrition and Health, Chinese Center for Disease Control and Prevention. All surveillance sites strictly followed the standardized protocols and data collection procedures. In addition, we collected information on a wide range of potential associated factors.

However, we should also acknowledge its limitations. First, the cross-sectional design of the study makes it difficult to evaluate temporal relationships of the associated factors with the prevalence of anaemia. Second, we could not exclude survivor bias, as only surviving infants were included. However, infants may have died from anaemia. Without the additional quantification of iron and inflammatory biomarkers (e.g., ferritin, sTfR, CRP, AGP), the analysis is unable to reveal how much of the anaemia burden is due to iron deficiency and how much would be expected to be responsive to iron interventions (e.g., distinguishing absolute iron deficiency vs. functional iron deficiency, the latter of which may not be responsive to simple oral iron interventions). Information such as this, in order to better understand the aetiology of the anaemia across the different regions, would be very valuable and desirable to obtain through future related studies. Third, the data were collected in 2013, which was relatively old. In the past decade, the YYB project was implemented in poor rural areas and improved the anaemia status of infants in these areas; in the future, based on new data, more socioeconomic and environmental factors should be used to analyse the prevalence of childhood anaemia.

## 5. Conclusions

The prevalence of anaemia among Chinese infants aged 0–23 months in 2013 was significantly high. China has experienced widening disparities in the prevalence of anaemia across regions, with a persistent disparity between urban and rural areas. Our results show that male sex, lower maternal educational level, younger maternal birth age, premature birth, LBW, breastfeeding and lack of compliance to MDD guidelines are significantly associated with higher odds of infants aged 0–23 months developing anaemia.

## Figures and Tables

**Figure 1 nutrients-13-01293-f001:**
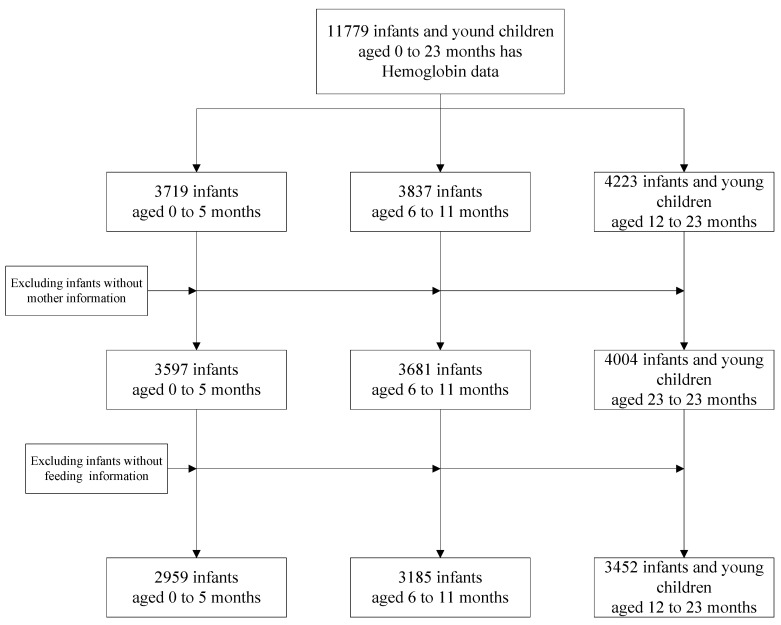
Flowchart of study procedures.

**Figure 2 nutrients-13-01293-f002:**
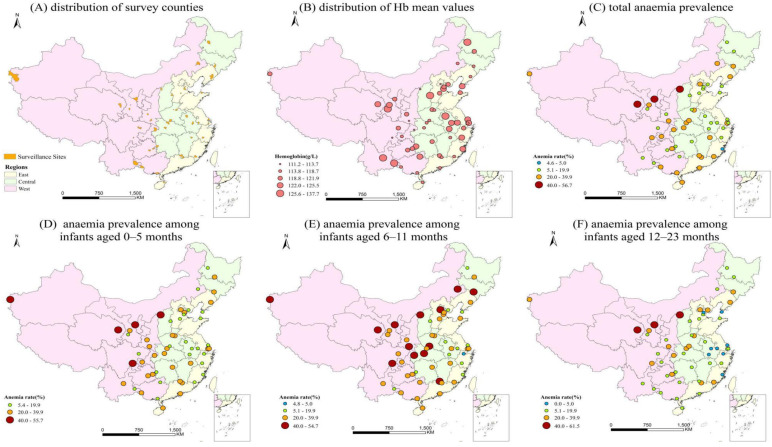
Spatial distribution map of Hb concentrations and anaemia prevalence. (**A**) Distribution of surveillance sites, (**B**) distribution of Hb mean values, (**C**) total anaemia prevalence, (**D**) anaemia prevalence among infants aged 0–5 months, (**E**) anaemia prevalence among infants aged 6–11 months and (**F**) anaemia prevalence among infants aged 12–23 months.

**Table 1 nutrients-13-01293-t001:** Sample status of infants and young children.

Characteristic	*N*	Percentage (%)
Age		
0–5 months old	2959	30.84
6–11 months old	3185	33.19
12–23 months old	3452	35.97
Sex		
Male	4921	51.28
Female	4675	48.72
Ethnicity		
Han	8260	86.08
Other	1156	12.05
Unknown	180	1.87
Premature birth		
Yes	1048	10.92
No	8548	89.08
LBW		
Yes	328	3.42
No	9268	96.58
Socioeconomic status		
BC	1967	20.50
SMC	2848	29.68
OC	3019	31.46
PC	1762	18.36
Regional division		
Eastern region	2993	31.19
Central region	3219	33.55
Western region	3384	35.26
Family status		
Family income		
<RMB35,000	7676	79.99
≥RMB35,000	1051	10.95
No answer	869	9.06
Toilet type		
Sanitary	6065	63.20
Unsanitary	3528	36.77
Unknown	3	0.03
Maternal status		
Education level		
High school and below	7507	78.24
College or above	2088	21.76
Unknown	1	0.00
Maternal age		
25 years old and under	2893	30.15
Over 25 years old	6703	69.85
Feeding status of 6–23-month-old infants		
Meet MDD		
Yes	3376	50.87
No	3261	49.13
Iron fortification		
Yes	2975	44.82
No	3662	55.18

LBW: low birth weight; BC: big city; SMC: small or medium-sized city; OC: ordinary county; PC: poverty-stricken county; MDD: minimum dietary diversity.

**Table 2 nutrients-13-01293-t002:** Haemoglobin status of infants stratified by sex, residential area and geographical region.

Months	0–5		6–11		12–23		Total	
	Mean	SE	Mean	SE	Mean	SE	Mean	SE
All	119.57 ^a^	0.83	117.18 ^a^	0.92	122.72 ^a^	0.81	120.67	0.79
Sex								
Male	119.18	0.83	116.83	0.90	122.59	0.82	120.44 *	0.77
Female	120.04	0.90	117.59	0.99	122.89	0.84	120.96 *	0.84
Residential area								
Urban	118.20 *	0.71	116.81	0.87	122.51	0.88	120.23 *	0.73
Rural	120.49 *	1.31	117.40	1.39	122.87	1.22	120.97 *	1.22
Geographical region								
Eastern	118.90 *	0.69	116.83 *	0.82	123.02	0.86	120.62 *	0.71
Central	120.46 *	1.99	118.43 *	2.14	122.52	1.68	121.03 *	1.75
Western	119.28 *	1.24	116.13 *	1.23	122.63	1.49	120.36 *	1.37

SE: standard error. Wilcoxon test results: ^a^ represented significant difference among different month age; * represented significant difference among sex, residential area or geographical region within one month age group or all infants.

**Table 3 nutrients-13-01293-t003:** Anaemia status of infants stratified by sex, residential area and geographical region.

Months	0–5		6–11		12–23		Total	
	*N* (%)	95% CI	*N* (%)	95% CI	*N* (%)	95% CI	*N* (%)	95% CI
All	944 (25.30)	23.87–26.74	1151 (30.64)	29.13–32.15	16.97	15.83–18.12	2798 (22.15)	21.36–22.94
Sex								
Male	511 (26.87)	24.82–28.92	640 (32.73)	30.60–34.86	381 (17.28)	15.70–18.87	1532 (23.14)	22.04–24.25
Female	433 (23.48)	21.48–25.48	511 (28.15)	26.03–30.27	322 (16.60)	14.93–18.26	1266 (20.96)	19.83–22.09
Residential area								
Urban	504 (26.05)	24.09–28.01	536 (27.99)	25.98–30.00	279 (14.17)	12.62–15.71	1319 (20.02)	18.95–21.09
Rural	440 (24.80)	22.80–26.82	615 (32.27)	30.17–34.37	424 (18.92)	17.30–20.55	1479 (23.57)	22.47–24.68
Geographical location								
Eastern	290 (20.88)	18.68–23.09	344 (25.17)	22.80–27.54	162 (11.12)	9.49–12.75	796 (16.67)	15.50–17.84
Central	289 (24.19)	21.71–26.66	364 (28.95)	26.39–31.50	249 (18.21)	16.16–20.27	902 (22.25)	20.88–23.63
Western	365 (30.97)	28.25–33.70	443 (37.96)	35.11–40.80	292 (21.42)	19.23–23.61	1100 (27.44)	25.93–28.95

CI: confidence interval.

**Table 4 nutrients-13-01293-t004:** Factors associated with anaemia among children aged 0–23 months, based on multivariate logistic regression analyses.

Influence Factor	Reference	0–5 Months		6–11 Months		12–23 Months	
		OR (95% CI)	*p*	OR (95% CI)	*p*	OR (95% CI)	*p*
Sex							
Female	Male	**0.80 (0.68–0.95)**	**0.01**	**0.77 (0.66–0.90)**	**0.00**	NS	
Premature							
Yes	No	**1.38 (1.06–1.81)**	**0.02**	**1.30 (1.01–1.67)**	**0.04**	NS	
Low birth weight							
Yes	No	**1.99 (1.28–3.09)**	**0.00**	NS		NS	
Breastfeeding							
Yes	No	**1.34 (1.04–1.72)**	**0.02**	**2.18 (1.84–2.60)**	**<0.01**	**2.17 (1.75–2.69)**	**<0.01**
Region							
Central	East	1.20(0.96–1.50)	0.10	1.21 (0.99–1.47)	0.07	**1.63 (1.25–2.13)**	**<0.01**
West	East	**1.66 (1.35–2.04)**	**<0.01**	**1.95 (1.60–2.37)**	**<0.01**	**2.11 (1.62–2.76)**	**<0.01**
Mother’s education level							
High	Low	**0.77 (0.63–0.95)**	**0.01**	**0.58 (0.47–0.72)**	**<0.01**	**0.70 (0.51–0.95)**	**0.02**
Residence areas							
SMC	BC	NS		NS		**1.80 (1.30–2.50)**	**<0.01**
OC	BC					1.25 (0.89–1.76)	0.20
PC	BC					1.31 (0.90–1.90)	0.15
MDD							
Yes	No	NA		NS		**0.60 (0.49–0.73)**	**<0.01**
Mother’s age							
Older	Younger	NS		NS		**0.70 (0.57–0.85)**	**<0.01**

NS: not significant; NA: not available. Note: the model of children aged 0–5 months adjusted for ethnicity, residence areas, maternal age, house toilet type, family income and Is formula-fed factors. The model of children aged 6–11 months adjusted for ethnicity, residence areas, maternal age, house toilet type, low birth weight, family income and iron-rich-food feeding factors. The model of children aged 12–23 months adjusted for ethnicity, residence areas, maternal age, house toilet type, low birth weight, family income and iron-rich-food feeding factors. The significant factors showed in bold.

## Data Availability

The data is not allowed to disclose according to the National Institute for Nutrition and Health, Chinese Center for Disease Control and Prevention.

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
