# Peer review of "Regional Differences in the Prevalence of Anaemia and Associated Risk Factors among Infants Aged 0–23 Months in China: China Nutrition and Health Surveillance"

_nutrients, 2021, doi:10.3390/nu13041293_

Round 1
Reviewer 1 Report
This is an interesting and carefully performed study. It is generally well written.
The survey that is the basis of the analysis presented collected data on the use of iron fortification. In reading the manuscript, it is unclear that iron fortification made a difference or not. When the authors speak of "food supplementation", do they mean meeting minimal dietary diversity standards, or are they referring to iron fortification, or both. Clarification this matter is required.
This paper is straightforward, carefully analyzed, and likely to contribute useful data to the understanding of the global problem of iron deficiency anemia as it applies to China. I think it is likely to generate a significant number of citations. The methodology is not novel but the findings are new and make a contribution.
Author Response
Response 1: thank you for your valuable comments. In discussion part, food supplementation meaned Meeting minimal dietary diversity(MDD) . We revised the statement to make it clear.
Reviewer 2 Report
This manuscript by Li et al is well constructed, describing a comprehensive analysis of factors associating with early childhood (0-23 months of age, N=9596) anemia across 55 locations within China, utilising data from the CNHS survey database from 2013. The authors investigate regional variation in anemia prevalence, and identify factors associated with anemia risk; these include male sex, poorer maternal education and younger maternal age, feeding practices, low birth weight and prematurity, prolonged breastfeeding, with differences in prevalence also described between regions of China. While none of these are surprising or unexpected in relation to previous studies of infant anemia from elsewhere, it is still valuable to describe these relationships across these regions of China through a large survey such as this.
I have the following minor comments and queries:
- Methods: How many infants were surveyed who did not provide Hb data? Is there any indication of why Hb measurement was not available, and whether this missing data would have been missing completely at random, or could have been skewed to certain groups within study populations potentially introducing bias?
- Table 2: could the authors add formal statistical testing for evidence of differences that the Hb concentrations between the categories mentioned (age, urban v rural etc) in Table 2?
- Related to Table 2: given the relatively large N in this study, I think it could be valuable for the authors to consider evaluating Hb concentration over the first 23 months of age with age as a continuous variable, besides the analysis with the 3 age categories, to model the age at which the nadir of Hb concentration occurs.
- Table 3: it would be helpful if the authors could represent the statistical analysis referred to in the text (lines 158-165) within the table.
- Figure 2: it would assist the reader to see a more prominent title for each panel on the Figure (especially to distinguish panels C, D, E and F). Please state if Hb concentrations are mean or median for Figure 2B.
- Figure 2B and 2C: two out of the three regions with severe (>40%) anaemia burden (in Qinghai and Ningxia provinces) appear to have relatively high Hb concentrations, which might seem unexpected. Could the authors confirm that this is correct, and comment – is there a much larger spread of Hb concentrations in these locations?
- There appears to be no Table 4 – please rename current Table 5 as new Table 4.
- Table 5: it would be preferable to report the OR (95% CI) for all data/variables for the reader to see, i.e. including those which are non-significant (NS). Those showing evidence of differences could be highlighted, e.g. in bold, or with asterisks.
- Table 5: please check the correspondence between the data reported in the text and that in the Table. P=0.00 should be corrected in the text on lines 196, 198, and 199. The data for city size (line 197-198) appears not to be given in Table 5 - this should be added.
- Discussion: line 217 – please reword to clarify “…and none had severe anemia”.
- Discussion: line 277-282 – This paragraph is currently quite difficult to interpret – it would be helpful to work on the wording of the paragraph. Presumably the suggestion is that increasing nutritional iron uptake should play a role in alleviating IDA? While this is not unreasonable, it would be important to point out that the data presented in the present manuscript is not able to reveal the extent to which iron deficiency (as opposed to other causes such as other nutritional deficiencies, inherited anemia or inflammation/infection) contributes to the burden of anemia. This is important in considering the likely effectiveness of iron interventions. If a significant proportion of anemia relates to inflammatory burden, for example, nutritional interventions may be ineffective in comparison to addressing inflammation (e.g. see discussion of similar issues within Pasricha et al, BMJ, 2018).
- Discussion: Limitations – in relation to the previous point, it would be worth highlighting that, without additional quantification of iron and inflammatory biomarkers (e.g. ferritin, sTfR, CRP, AGP), the analysis is unable to reveal how much of the anemia burden is due to iron deficiency and how much would be expected to be responsive to iron interventions (e.g. distinguishing absolute iron deficiency vs functional iron deficiency, the latter of which may not be responsive to simple oral iron interventions). Information such as this, in order to better understand the etiology of the anemia across the different regions, would be very valuable and desirable to obtain through future related studies.
- Discussion: Limitations – given that the survey was carried out in 2013, the authors could comment on how well the data might be expected to reflect the situation in 2021 – approaching a decade later. Have any interventions been introduced or assessed in the meantime for example?
- Conclusion: Line 299 – male sex is associated with higher anemia risk, not female sex. Please revise.
